



# The development and validation of a global 1/32° surface wave-tide-circulation coupled ocean model: FIO-COM32

Bin Xiao[1,2,3], Fangli Qiao[1,2,3*], Qi Shu[1,2,3], Xunqiang Yin[1,2,3], Guansuo Wang[1,2,3], Shihong Wang[1,2,3]

[1]First Institute of Oceanography, and Key Laboratory of Marine Science and Numerical Modeling, Ministry of Natural Resources, Qingdao 266061, China
[2]Laboratory for Regional Oceanography and Numerical Modeling, Pilot National Laboratory for Marine Science and Technology, Qingdao 266237, China
[3]Shandong Key Laboratory of Marine Science and Numerical Modeling, Qingdao 266061, China

*Correspondence to*: Fangli Qiao (qiaofl@fio.org.cn)

**Abstract.** Model resolution and the included physical processes are two of the most important factors that determine the realism of the ocean model simulations. In this study, a new global surface wave-tide-circulation coupled ocean model FIO-COM32 with resolution of 1/32°×1/32° is developed and validated. Promotion of the horizontal resolution from 1/10° to 1/32° leads to significant improvements of the simulations of surface eddy kinetic energy (EKE), fine structures of sub-mesoscale to mesoscale movements and the accuracy of simulated global tide. The non-breaking surface wave-induced mixing (Bv) is proved to be an important contributor that improves the agreement of the simulated summer mixed layer depth (MLD) of the model and the Argo observations even with high horizontal resolution of 1/32°, the mean error of the simulated mid-latitude summer MLD is reduced from -4.8 m in numerical experiment without Bv to -0.6 m in experiment with Bv. With the global tide is included, the global distributions of internal tide can be explicitly simulated in this new model and is comparable to the satellite observations. Comparisons using Jason3 along-track sea surface height (SSH) wave-number spectral slopes of mesoscale ranges show that internal tide induced SSH undulations is a key factor contributing to the substantially improved agreement of model and satellite observations in the low latitude and low EKE regions. For ocean model community, surface wave, tidal current and ocean circulation have been separating into different streams for more than half century. It should be the time to merge these streams for new generation ocean model development.

## 1 Introduction

As the computing power has been increasing rapidly, about one order higher with each five years, the state-of-art total computing ability of the modern global ocean numerical models is becoming enormously high, leading to the recent achievements of global high resolution ocean models. The definition of "high resolution" of current stage global ocean models may refer to these with horizontal resolution range from 1 to 5 km, which are well beyond the mesoscale resolving threshold in most of open ocean (Hallberg, 2013). Further improved resolution has significant impact on the simulated





eddies activities (Thoppil et al., 2011; Sasaki and Klein, 2012; Biri et al., 2016; Ajayi et al., 2020), the vertical mass and buoyancy fluxes (Capet et al., 2016; Su et al., 2018; Dong et al., 2020), and even the representation of large scale circulations (Lévy et al., 2010; Chassignet and Xu, 2017). These high resolution models are not only helpful to gain

knowledge on mesoscale, submeso-scale, and mixed layer eddies, but also very useful in evaluating the satellite products of both present-day (Amores et al., 2018) and future generations, such as SWOT project. Table 1 summarizes recent developments of the global high resolution ocean models, not only the model horizontal resolution is significantly increased, which now ranges from 1/20° to 1/48°, but also the ocean model physical processes are making notable progresses, for example, the HYCOM25 and LLC4320 are global tide-circulation coupled and FIO-COM32 is global surface wave-tide-

circulation coupled.

**Table 1 Recent developments of the global high resolution ocean models**

| Global high resolution models | Ocean model codes | Horizontal resolution | References |
|---|---|---|---|
| HYCOM25 | HYCOM | 1/25° | Savage et al., 2017ab; Arbic et al., 2018 |
| LLC4320 | MITgcm | 1/48° | Rocha et al., 2016 |
| ORCA36 | NEMO4 | 1/36° | https://github.com/immerse-project/ ORCA36-demonstrator |
| LICOM3-HIP | LICOM3 | 1/20° | Wang et al., 2020 |
| FIO-COM32 | FIO-COM-HR | 1/32° | This paper |

Improving the representation of the physical processes of the ocean model has been the most fundamental aspect for ocean model development. Since the establishment of the first ocean general circulation model (OGCM, Bryan and Cox,

1967), surface wave models, ocean tide models, ocean internal wave models and OGCMs have been separated into different streams (Mellor and Blumberg, 2004). And the most uncertainty term of ocean turbulence in all OGCMs is far from observation. As a result, the vertical structures of ocean temperature and salinity can not be accurately simulated and predicted. For example, the simulated mixed layer depth (MLD) in the upper ocean is too shallow, and the sea surface temperature (SST) is overheating in summer in nearly all OGCMs. Qiao et al (2004) found that the non-breaking surface

wave-induced mixing (Bv) plays a dominant role for the mixing in the upper ocean, analytically expressed Bv as the function of wave number spectrum which can be calculated from a wave model, and developed the Bv theory (Qiao et al., 2004, 2016). The ocean surface wave-circulation coupling has been becoming a most important direction of the ocean model development. Bv has been widely adopted in a series of different ocean and climate models and all showed dramatic improvements (Qiao et al., 2004; Shu et al., 2010; Song et al., 2012; Fan and Griffies, 2014; Wang et al., 2019). Bv is

particularly effective in remedy the shallow biases of simulated summer MLD and overestimated SST in summer. Additional





two points are surprising: Firstly, even closing the traditional vertical turbulence schemes, only Bv is able to reproduce proper simulations of the global ocean which indicates Bv plays dominant role for the vertical mixing in the upper ocean (Qiao and Huang, 2012); Secondly, Bv can shallow the MLD in winter in climate model which is also an improvement for climate models (Chen et al., 2019). Although the effect of Bv has been comprehensively verified in many course resolution

OGCMs. Taking the significantly different behaviors of course and high resolution models into consideration, it is still necessary to validate the effect of Bv in the high resolution models forced by real-time atmosphere data with more observations.

The ocean tide has long been recognized as a fundamental aspect regulating the hydrodynamic environments in shallow regions (e.g. Simpson and Hunter, 1974; Garrett and Loder, 1981; Holt and Umlauf, 2008; Lin et al., 2020), thus the ocean

tide is often included in the regional ocean models. The tidal mixing controls the formation of thermal fronts in coastal regions, and generates upwelling (Lü et al., 2006, 2008, 2010). The regional surface wave-tide-circulation coupled ocean model of China Seas has shown excellent performance (Xia et al., 2006), and has been applied in operational ocean forecasting systems (Wang et al; 2016). In recent years, the high resolution global ocean model forced by a realistic atmospheric data begins to include ocean tide explicitly (Arbic, et al. 2010, 2012, 2018; Rocha et al., 2016). The benefit of

including ocean tide in the eddy-resolving resolution global OGCM is that the global distribution of internal tide field together with mesoscale eddies can be resolved concurrently (e.g. Arbic et al., 2010, 2012, 2018; Buijsman et al., 2015; Shriver et al. 2012; Ansong et al., 2018; Timko et al., 2019). In addition, inclusion of ocean tide in a global 1/48° ocean model (MITgcm llc4320) leads to more realistic representation of the unbalanced and ageostrophic motions (Rocha et al., 2016).

In order to examine whether the high resolution OGCM can faithfully reproduce the ocean state, the comparison of wave-number/frequency spectra with observations is widely adopted (e.g. Sasaki and Klein, 2012; Richman et al., 2012; Chassignet and Xu, 2017, hereafter CX17; Savage et al., 2017ab; Biri et al., 2018). So, the wave number spectral slope in the 70–250 km mesoscale range becomes an important criterion for OGCMs' validation. Sasaki and Klein (2012), hereafter SK12, note that as the horizontal resolution improved to 1/30°, the wave number spectral slope in the 70–250 km mesoscale

range of the model shows quite inconsistent patterns with that of satellite observations. The satellite observations show strong latitudinal variability (Xu and Fu, 2012), in high latitude and high eddy kinetic energy (EKE) regions the slopes range between -5 to -4 which is generally consistent with the theory of mesoscale turbulence, however, the slopes are much flatter in tropical and low EKE regions which is substantially different with the theory of mesoscale turbulence. While in the high resolution models such as SK12 and CX17, the slopes range between -5 to -4 in both low to high latitude regions. More

importantly, Chassignet and Xu (2021), hereafter CX21, has shown that as the tide is included in their 1/50° Atlantic regional ocean model, the resolved internal tide sea surface height (SSH) signals are the main reason that could explain the inconsistency between model and satellite observations. Although ocean tide is included in regional OGCMs, there is still few tries to include tide into global OGCMs.





---

This paper aims to answer the following two questions through establishing a global 1/32° surface wave-tide-circulation
coupled ocean model FIO-COM32: What is effect of Bv in high resolution OGCM? Inspired by the results of an Atlantic
regional high resolution ocean model of CX21, what is the tidal effect in a global 1/32° high-resolution OGCM on
simulating the wave number spectral slopes?

This paper is organized as follows. Section 2 describes the model configurations and design of numerical experiments. In
section 3.1, we present the results of basic aspects of the new global 1/32° surface wave-tide-circulation coupled ocean
model, and by comparing with a previous global 1/10° model we illustrate the effects of increased horizontal resolution.
Section 3.2 shows effects of ocean surface wave and tide coupling in the global 1/32° ocean model. Section 4 makes
summary and discussions.

## 2. Model description and numerical experiments design

### 2.1 Model description

The FIO-COM consists of a global ocean circulation model, a sea ice model, and a global ocean wave model. The
ocean component is based on the Modular Ocean Model 5 (Griffies, 2012), the sea ice component is based on the Sea Ice
Simulator (Winton, 2000), and the surface wave component is based on the MASNUM (laboratory of MArine Sciences and
NUmerical Modeling, MNR, China) ocean wave model (Yang et al., 2005; Qiao et al., 2016). Currently, the variable
exchanged between the ocean wave and the ocean circulation model is Bv. Bv is calculated in the ocean wave model
following Qiao et al., (2004), as in equation (1), $S(k)$ represents the surface wave-number spectrum, $k$ is the surface wave-
number. As Bv is incorporated into the vertical mixing schemes of the ocean circulation model, α is a tune-able parameter
for practical purposes, we set it as 0.3 in this paper.

$$B_v = \alpha \int_k \ S(k)e^{-2|k|z}dk \cdot \frac{\partial}{\partial z}[\int_k \ \omega^2 S \ (k) \ e^{-2|k|z}dk]^{\frac{1}{2}} \qquad (1)$$

The variable exchanged between the ocean wave and the sea ice models is sea ice concentration (SIC, from sea ice
model to ocean wave model). The SIC is used to calculate time varying masks of the ocean wave model, the model domain
that has SIC exceeding 15% is masked out as land during the ocean wave model integration.

According to different research purposes, there are two different coupling strategies for the wave model component:
Firstly, in the lower resolution numerical experiments, where the computational costs are not expensive, the wave model
component is coupled on-line with the ocean circulation and sea ice model components. The real-time on-line data
exchanges are achieved based on the subroutine version of MASNUM wave model. In this way, the wave model component
could be coupled with ocean circulation and ice components through direct calling of the MASNUM wave model as
subroutines in the ocean circulation model codes. The ocean wave, ocean circulation, and sea ice components share the same
model grids. Secondly, in the numerical experiments of high computational costs, and with research focus on the ocean
dynamics, the wave component can be turned off to save computational resources (the computational cost of ocean-ice





model is about half that of wave-ocean-ice model). In this configuration, the wave induced mixing coefficients which were saved in data files previously are read into the ocean model.

   The horizontal grid of FIO-COM32 is a tri-polar grid (Murray, 1996) with a horizontal resolution of $1/32° \times 1/32°$. The model covers the entire global ocean, with a latitudinal coverage of 82°S to 90°N and a longitudinal coverage of 280°W to 80°E respectively. Vertically, z* coordinate (Adcroft and Campin, 2004) is adopted and the thickness is 2 m at surface and

increases to 367 m at the bottom gradually. The vertical grid has 54 or 57 levels depend on whether or not ocean tide is introduced. In the numerical experiments without ocean tide, the maximum depth is 5500 m, while the ocean tide is explicitly included the maximum depth extends to 7000 m by adding additional 3 bottom levels. The maximum total grid size is $11520 \times 5504 \times 57$. Model topography is derived from GEBCO (IHO-IOC, 2018) and is smoothed by applying a radial filter (Arbic et al., 2004). Model bottom cells are set as partial cells (Adcroft et al., 1997). The horizontal mixing scheme is a

bi-harmonic operator with diffusive velocity of 1.96 cm/s for momentum and 0.65 cm/s for tracers respectively. The vertical mixing scheme is the KPP scheme (Large et al., 1994) with non-breaking wave induced mixing Bv. The background vertical viscosity and diffusivity are set as $1.0 \times 10^{-4}$ $m^2s^{-1}$ and $3.0 \times 10^{-5}$ $m^2s^{-1}$ respectively. Bv is incorporated into both vertical diffusivity and viscosity. Neither sea surface temperature nor salinity is restored to observed data. The bottom drag is quadratic with a coefficient of $2.5 \times 10^{-3}$.

Ocean tide is explicitly included by introducing eight main tidal generating potentials including $M_2$, $S_2$, $N_2$, $K_2$, $K_1$, $O_1$, $P_1$, and $Q_1$ in the momentum equations following Schiller and Fiedler (2007). The treatment of self-attraction and loading (SAL) is a simple scalar approximation following Arbic et al. (2010), and the scalar alpha is set to 0.93. The SAL is applied only on the tidal elevations, thus a 25-hour running average time filter is adopted to get the slow-varying sea surface height related with large scale ocean circulation. A topographic drag scheme (Jayne and St. Laurent, 2001) is introduced in the

barotropic momentum equation, and this scheme is enabled in the barotropic experiments but closed in the baroclinic experiments for that a large portion of the barotropic to baroclinic tidal energy conversion can be explicitly resolved in the baroclinic experiments.

   The initial conditions including sea water temperature, salinity, velocity and sea surface height (SSH) are interpolated from the outcomes of a global operational ocean forecasting system based on FIO-COM with horizontal resolution of 1/10° (Q

iao et al., 2019; Shi et al., 2018; Sun et al., 2020). The atmospheric forcing is from the Global Forecast System (GFS) of National Centers for Environmental Prediction (NCEP) (https://www.nco.ncep.noaa.gov/pmb/products/gfs/) with a horizontal resolution of 1/4°. The ocean surface heat and momentum fluxes are calculated via the bulk formula (Large and Yeager, 2004), and the wind stress is calculated using relative wind.

## 2.2 Numerical experiments design

In order to investigate to what extent the surface wave-tide-circulation coupled ocean modelling framework contribute in the newly established FIO-COM32 model, and taking the expensive computational cost into consideration, two numerical experiments are designed. In EXP1, only OGCM is active neither ocean surface wave nor tide is included, and the simulated



period is from 1 June 2016 to 31 December 2019. In EXP2, wave-tide-circulation fully coupling is enabled which means both Bv and tidal currents are activated. Since the on-line coupled surface wave model would almost double the needed

computational resources compared with the ocean-only counterpart, in this paper the 1/32° numerical experiments reads in the Bv data which were calculated from a global 1/4° on-line coupled FIO-COM for the exactly same time period. Here we mainly focus on the effect of Bv in the new 1/32° ocean model, and due to the large-scale characteristics of the surface wave simulations and the extremely expensive computational costs, we avoid to run a global 1/32° on-line coupled ocean-wave model directly.

EXP1 starts from outcomes of the global 1/10° FIO-COM forecast system of 1 June 2016. EXP2 branches from EXP1 on 1 July 2017. The data analysis focus on period from 1 January 2018 to 31 December 2019. The model output frequency of three-dimensional variables is daily, and the output frequency of SSH and steric SSH is hourly. An additional experiment with horizontal resolution of 1/10° is also conducted to investigate the effects of different model resolution. The settings of the 1/10° model are kept identical with that of EXP1, except for the different horizontal resolutions.

Another two numerical experiments, OGCM+Bv and OGCM+tide, should be conducted. Considering the high computational costs, we only do EXP1 and EXP2.

## 3 Results

### 3.1 Effects of increased horizontal resolution

In this section, the surface EKE and SSH simulated by EXP1 are compared with a 1/10° model results to investigate

the effects of increased horizontal resolution.

One of the most noteworthy facts is that the eddy kinetic energy increases significantly as the model horizontal resolution increased, as CX17 noted that this increase is due to smaller effective horizontal viscosity and more active mesoscale and submeso-scale motions of higher resolution models. The simulated surface EKE and satellite observations are compared here to investigate the effect of improved horizontal resolution in FIO-COM32. The surface EKE is calculated as

follows.

$$EKE = \frac{1}{2}\langle u'^2 + v'^2 \rangle \quad (2)$$

where $u'$ and $v'$ are anomalous zonal and meridional components of surface geostrophic velocity, respectively, which are calculated from sea level anomalies referenced to mean sea level of year of 2018 and 2019. The bracket indicates time averaging of model year of 2018 and 2019. Satellite derived surface EKE are calculated using absolute dynamic topography

data of Ssalto/Duacs altimeter products which are produced by the Copernicus Marine and Environment Monitoring service (CMEMS, http://marine.copernicus.eu). Generally, increasing model resolution from 1/10° to 1/32° significantly improves the agreement of simulated EKE and satellite observations (Figure 1). Compared with the 1/10° model, the 1/32° model shows much enhanced EKE almost everywhere in the global ocean, especially in the regions of sub-tropic, mid-latitude, and





western boundary current systems, where surface EKE in the 1/10° model is too weak against satellite data. This indicates

the horizontal resolution of 1/10° may be insufficient to properly resolve mesoscale and sub-mesoscale motions, while the

simulations of the 1/32° model are much improved. For the Kuroshio and Gulfstream extensions, not only the EKE amplitude

is enhanced but also their spatial distribution structures such as the separation point and the extension range agree with the

satellite observations much better in the 1/32° model. The calculated global mean EKE time series (Figure 1d) also show

improved agreement of the 1/32° model with the satellite observations, while the 1/32° model has higher global mean EKE

values. The calculated root mean square error (RMSE) of the two-year-averaged EKE between model simulations and

satellite observations are decreased from 63.5 $cm^2 s^{-2}$ of the 1/10° model to 31.7 $cm^2 s^{-2}$ of the 1/32° model. The RMSE

is calculated for regions that have water depth exceeds 1000 m and locate between 65° S and 65° N.

Mean SSH and SSH standard deviation (STD) of the 1/10° and 1/32° model are compared in Figure 2. The large scale

circulation patterns are reasonably simulated in both the 1/10° and 1/32° models (Figures 2a, c, e). While for the detailed

simulation structures, such as pathways and separations of the west boundary currents, the 1/32° model shows significant

improvements over that of the 1/10° model. As the SSH STD comparisons show, the 1/32° model is more energetic than the

1/10° model in the well-known mesoscale active regions (Figures 2b, d, f), which has already been discussed in the EKE

analysis. In the Kuroshio region, the 1/10° model shows over concentrated energy near the separating point and the Kuroshio

extension is too weak to penetrate into the interior of Pacific Ocean, while these characteristics are much improved in the

1/32° model.

Figures 3 and 4 show the simulated relative vorticity of sea surface current of the 1/10° and 1/32° models. They clearly

show that as the horizontal resolution increases the model can resolve much more sub-mesoscale and mesoscale motions.

The 1/32° model also shows obviously more significant seasonal variations over the 1/10° model, as have been described in

previous studies (Sasaki et al., 2017; CX17; Dong et al., 2020). In the higher resolution model the small-scale eddy activities

is greatly enhanced during winter compared to that of summer, while this is not as obvious in 1/10° model.

The accuracy of simulated global barotropic tide is quite sensitive to the model resolution (Egbert et al., 2004), and

reasonable global tide accuracy is an important prerequisite that tide and ocean general circulation could be coupled together.

Here we show the effects of improving horizontal resolutions from 1/10° to 1/32° on the global barotropic tide simulations of

FIO-COM.

Two global barotropic tide models with horizontal resolution of 1/10° and 1/32° are compared. Ocean temperature and

salinity of the global barotropic tide models are set as uniform values, and are kept as constants during model integration.

The parameterized topography drag scheme is needed in the barotropic tide models for that the energy transfer mechanism

from barotropic to baroclinic tide is missing. A topography drag scheme (Jayne and St. Laurent, 2002) is incorporated into

the momentum equation following previous works (Jayne and St. Laurent 2002; Egbert et al., 2004; Arbic et al., 2004), the


drag coefficient is best tuned for both numerical experiments. A scalar approximation method (Arbic et al., 2010) is adopted

for SAL effect. The topography of both models is from GEBCO as have been described in model description of Section 3.

The model topography in this paper is different from our previous work (Xiao et al., 2016) in which the treatment of ice shelf

topography of Antarctica is more realistic by using the water column thickness of BEDMAP2 (Fretwell et al., 2013), and this

would yield smaller RMSE.

For the 1/10° model, the tidal amplitude is weaker than TPXO9 in some regions such as west of Panama and north-east

Pacific, while the tidal amplitude is stronger than TPXO9 in the Labrador, Okhotsk and Andaman seas (Figure 5). These

biases are obviously reduced in the 1/32° model, which indicates the tidal dissipation in the regions of complex terrain is

better resolved as the horizontal resolution is increased. The RMSE distributions show that the tidal accuracy of 1/32° model

is improved almost everywhere over that of 1/10° model. The statistic shows that for the $M_2$ constituent, the global averaged

tidal elevation RMSE is decreased from 9.65 cm to 8.06 cm, reduced by 16.4%. Here, the tidal elevation RMSE is calculated

for regions have water depth exceeding 1000 m and locate between 65° S and 65° N following previous work (Arbic et al.,

2004; Egbert et al., 2004). We believe that the global tide accuracy is reasonable to support the results in this paper.

### 3.2 Effects of surface wave-tide-circulation coupling

Prior to examining the effects of surface wave-induced mixing in the newly established FIO-COM32 model, the

accuracy of the ocean wave model is evaluated. Figure 6 shows the comparisons of Jason3 and model simulated significant

wave height (SWH). From the comparison, the model results are interpolated onto the satellite ground tracks. Both the

distributions and seasonal cycle can be well reproduced by the global 1/4° on-line coupled FIO-COM. The RMSE of the

simulated SWH against Jason3 along track geophysical data records is calculated. The Jason3 data are produced and

distributed by Archiving, Validation, and Interpolation of Satellite Oceanographic data (AVISO,

http://www.aviso.altimetry.fr/en/home.html). We use the low-pass filtered Jason3 data for the calculations, the global mean

RMSE of the simulated SWH is 0.57 m.

The ocean upper mixed layer is a crucial "gate" locating between ocean interior and atmosphere. The mixed layer depth

(MLD) determines the heat and momentum content of the upper ocean boundary layer, and is a key factor that reflects the

ability of an ocean model in modelling the upper ocean. Figure 7 shows the derived summer MLD based on Argo

observations (a), EXP1 (b) and EXP2 (c) simulations. The derived summer is the mean of July, August and September (JAS)

for the northern hemisphere, and the mean of January, February and March (JFM) for the southern hemisphere, respectively.

The MLD is defined as the depth at which potential density increases by 0.125 kg/m³ from the surface values. The model

results are interpolated onto the Argo profiles in both space and time.

Both EXP1 and EXP2 are able to reproduce the general patterns of summer MLD distributions. However, the MLD of

EXP1 without Bv is shallower than that of Argo observations in most of the ex-tropical regions (Figure 7d). These shallow

biases are significantly alleviated in EXP2 (Figure 7e), which is due to the inclusion of Bv as previous works. The



comparison of the zonal mean MLD shows great improvement of EXP2 over EXP1. EXP2 fits quite well with the observations except in the regions of Antarctic Circumpolar Current. The mean error of MLD in mid-latitude (10º-40º N/S) is decreased from -4.8 m (EXP1) to -0.6 m (EXP2). Although the resolution is increased to 1/32°, the Bv is still an important contributor that improves the summer MLD simulations.


As model resolution is increased and global tide is explicitly included in EXP2, the model is able to simulate global field of internal tide (IT), which is activated when oscillating tidal currents flow over rough topography. Since a large portion of barotropic to baroclinic energy transfer can be explicitly resolved in EXP2, we disable the parameterized topography drag which has been used in the barotropic model. Figure 8 compares the global internal tide fields of satellite

based data MOIST (Multivariate Inversion of Ocean Surface Topography) (Ubelmann et al., 2021) and model simulations. We apply the radial filter with the radius of 4º as high-pass filter to remove the barotropic tide signals in the open ocean in the model SSH output. After that, harmonic analysis of 1-year high-passed SSH time series is performed to obtain the simulated $M_2$ internal tide amplitude. As shown in Figure 8, for both the $M_2$ and $K_1$ constituents the MOIST and EXP2 agree well with each other in the positions of the IT "hot spot" generation sites, and the long-range propagation patterns. For

example, the well-known generation site of $M_2$ IT near the Hawaii, and world's strongest $K_1$ IT near Luzon straits are well reproduced in the model. The model simulations tend to be more energetic than the MOIST data, and whether this bias is due to the mesoscale contamination of the satellite data in the MOIST or insufficient IT related energy dissipation in present model is remain to be explored in the future.

Previous studies (SK12, CX17) have reported that the high-resolution model simulated SSH wave-number spectral slope in the 70–250 km mesoscale range is quite different from that of satellite observations. In the models, the slopes range between -5 to -4 in both low to high latitude regions. While the satellite observations show strong latitudinal variability, with much flatter slope of about -1 in tropical ocean to about -5 to -4 in high latitude and high EKE regions. CX21 shows internal tide induced SSH signals is a critical factor that influences the wave-number spectral slopes. With tide included, the wave-

number spectral slopes in the 70–250 km mesoscale range of their 1/50° Atlantic regional ocean model show much better agreement with that of satellite observations than the model results without tide. Figure 8 shows that the internal tide induced sea surface undulations with spatial scale of tens to hundreds kilometres can reach several centimetres, so it is needed to explore the different responses of wave-number spectral slopes in the EXP1 and EXP2.

The SSH wave-number spectral slope of 70-250 km range is calculated as follows: Firstly, the model SSH data are

interpolated onto Jason3 ground tracks. Secondly, the spectral slopes are calculated on a 1° x 1° grid for the global ocean. For each grid point, tracks falling into a 10° x 10° super-domain centered at the grid point are taken for spectral calculation. In addition, it also requires sub-tracks are longer than 500 km and have 90% or more good quality data. Prior to spectral analysis, the trend of each sub-track is removed, and a Hanning window is applied. The spectra is obtained by averaging all





the SSH spectra obtained via a fast Fourier transform (FFT). Finally, the spectral slope is calculated in the 70–250 km
mesoscale range, which is in line with Xu and Fu (2012).

      Figures 9a, b and c show SSH wave-number spectral slope of 70-250 km range of Jason3 along track filtered data,
EXP1 and EXP2. Figure 9a resembles to the results of Xu and Fu (2012), which shows strong latitudinal variability of wave-
number spectral slopes of the satellite observations. Figure 9b shows the wave-number spectral slopes of EXP1 has much
weaker latitudinal variability, which resembles to that of SK12 and CX17. While the wave-number spectral slopes of EXP2
show substantially improved agreement with that of satellite observations shown in Figure 9a. The flattened wave-number
spectral slopes of EXP2 in low EKE regions result from internal tide induced SSH undulations which can be clearly
observed in SSH snapshots (Figures 9g and h). These internal tide signals have spatial scale of tens to hundreds of kilometers,
and become nontrivial where the background eddy and circulation field is relatively weak. Figures 9d, e and f show wave-
number spectra of three representative sites. In the high latitude and high EKE regions the spectra is less affected by the
internal tide signals. However in the low latitude and low EKE regions, the presence of internal tide substantially alters the
spectra and yield much more realistic simulations. The textures of SSH of EXP1 and EXP 2 shown in Figures 9g and h
manifest a lot of differences which mainly result from the internal tide signals, and the EXP2 would "feel" much closer to
the reality. So for high resolution global ocean models, tide-circulation coupling is important for more detailed and realistic
SSH simulations.

**4. Summary and discussions**

      In this paper, a global 1/32° surface wave-tide-circulation fully coupled ocean model is developed and validated.
Increasing of model horizontal resolution from 1/10° to 1/32° leads to great improvements of the simulated surface EKE,
fine structures of sub-mesoscale to mesoscale movements, and global tide accuracy is also improved. The RMSE of EKE
between model simulations and satellite observations are decreased from 63.5 $cm^2s^{-2}$ of the 1/10° model to 31.7 $cm^2s^{-2}$
of the 1/32° model.  The global barotropic $M_2$ RMSE is decreased from 9.65 cm of the 1/10° model to 8.06 cm of the 1/32°
model. Although the resolution is increased to 1/32° , the non-breaking wave induced mixing (Bv) is still a key factor in
improving the simulated summer MLD against the Argo observations, with the mean error of mid-latitude summer MLD
reduced from -4.8 m in numerical experiment without Bv to -0.6 m with Bv. Internal tide can be explicitly simulated in this
new model, the global comparisons of along-track SSH wave-number spectral slopes in 70-250 km shows that internal tide
induced SSH undulations is a critical factor that contribute to substantially improved agreement between the global 1/32°
model and satellite.

      In our following works, the effects of ocean current to the wave model is also need to be taken into consideration,
which has shown to have important impact on the simulated surface wave properties of down to 10-100 km scales (Ardhuin
et al., 2017). Since the internal tide can be resolved in the FIO-COM32 model, the more robust representation of energy
dissipation of internal tide propagation is still an open topic that need to be addressed in the future. At the same time, more

proper representation of the unresolved barotropic to baroclinic tidal conversion needs to be included in the future. More realistic treatments of SAL will benefit the accuracy of global barotropic tide models (Arbic et al., 2004).

Better prediction of ocean status is the inexhaustible power of ocean model community. As ocean, especially the upper ocean plays dominant role in climate system, the ocean model improvement can shed light on new generation climate model development. Surface wave, tide and ocean circulation are often simulated by different numerical models separately, here we clearly show surface wave-tide-circulation fully coupled model can dramatically improve our simulations, and should be the main stream of future ocean model development.

**Code and data availability**

The exact version of the model, input data used to produce the results used in this paper and data to produce the plots are
archived on Zenodo (https://doi.org/10.5281/zenodo.6221095), the minimal scripts to compile and run the model can be accessed on Github (https://github.com/mom-ocean/MOM5). The atmosphere forcing data is downloaded from https://www.nco.ncep.noaa.gov/pmb/products/gfs/.

**Author contributions**

BX was responsible for model development and drafted the manuscript; FQ designed the road map of this research and
supervised the paper writing. All other authors contributed through discussion, data analysis and the paper writing.

**Competing interests**

The authors declare that they have no conflict of interest.

**Acknowledgements**

This research is jointly supported by the National Natural Science Foundation of China under Grant 41821004 and the
Marine S&T Fund of Shandong Province for Pilot National Laboratory for Marine Science and Technology(Qingdao) (No. 2018SDKJ0106-1).

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

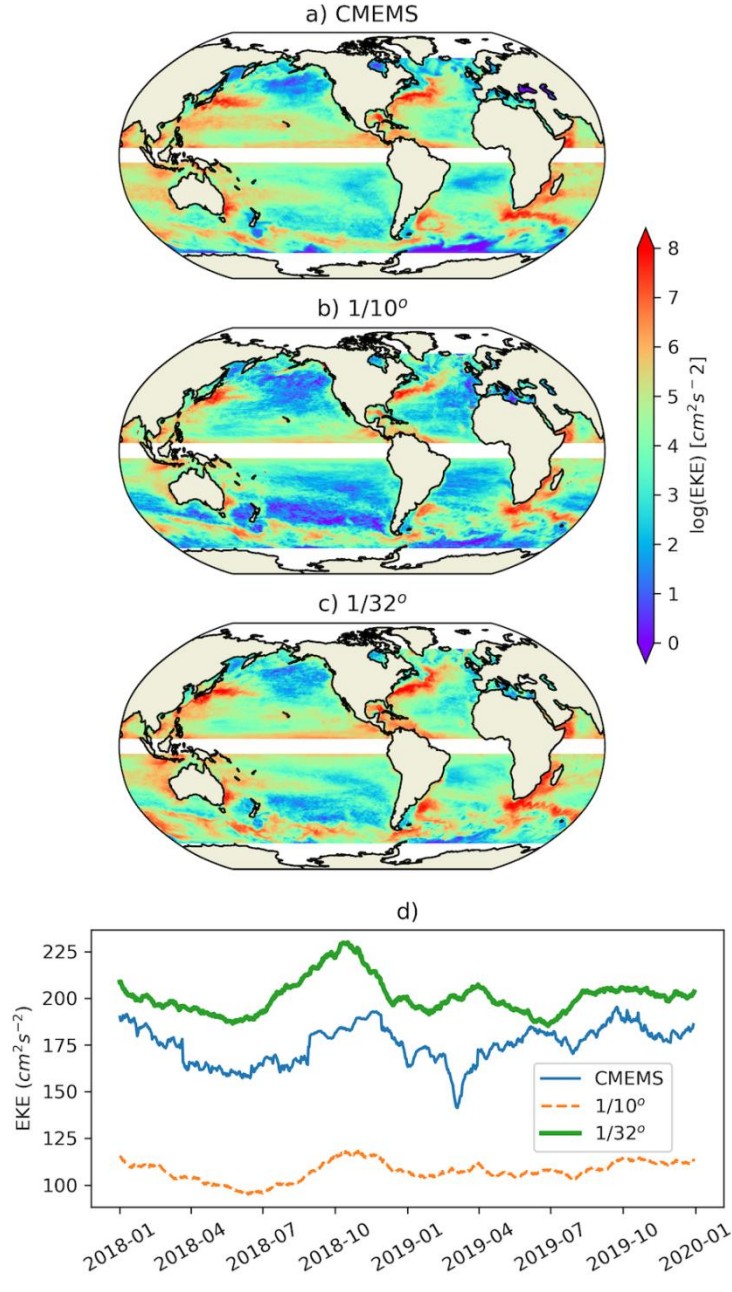

**Figure 1 Eddy kinetic energy (EKE) of CMEMS all satellite merged grided data (a), FIO-COM 1/10° (b), 1/32° (c) model results and the corresponding time series (d).**


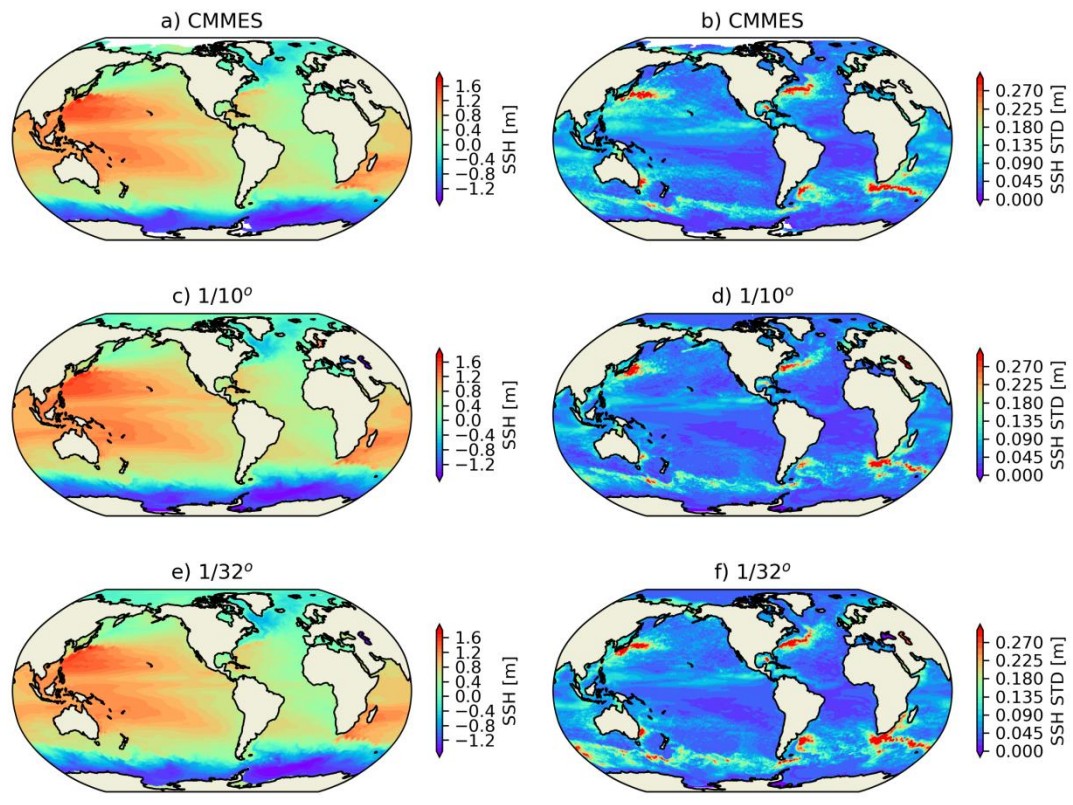

**Figure 2 Mean SSH (first column) and SSH STD (second column) based on altimeter observations of CMEMS (a, b), FIO-COM 1/10° (c, d) and 1/32° (e, f) simulations.**





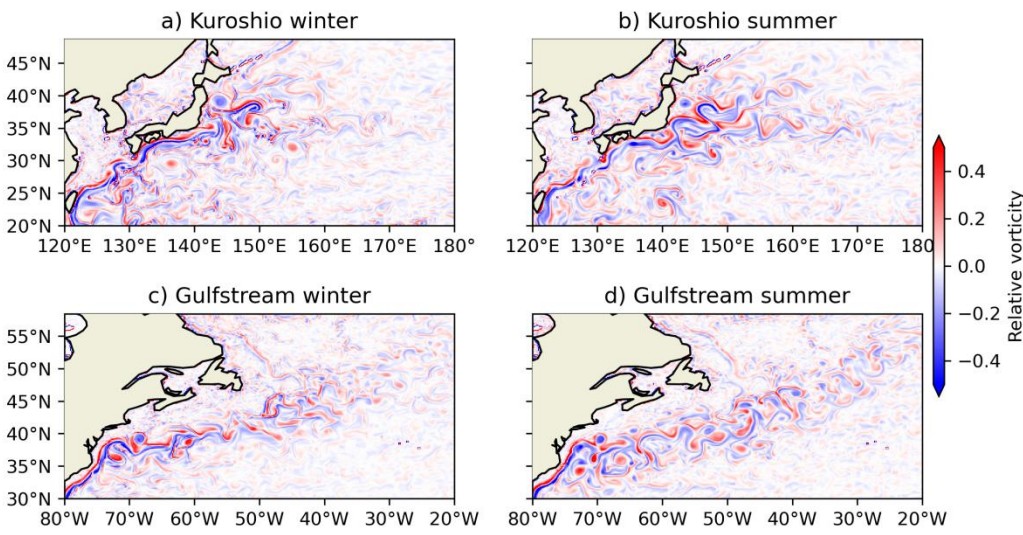

**Figure 3 Relative vorticity ( $\dfrac{\zeta}{f}$ ) of sea surface current of 1/10° model, snapshots of winter (1 March 2019) and summer (1 September 2019) of Kuroshio region (a, b), and that of Gulfstream region (c, d).**

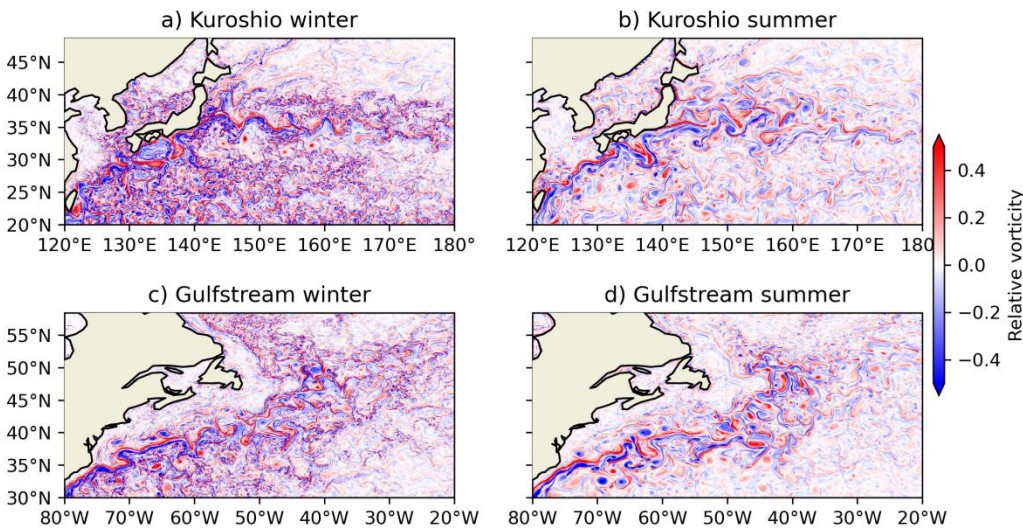

495                Figure 4 Same as Figure 3 but for 1/32° model.



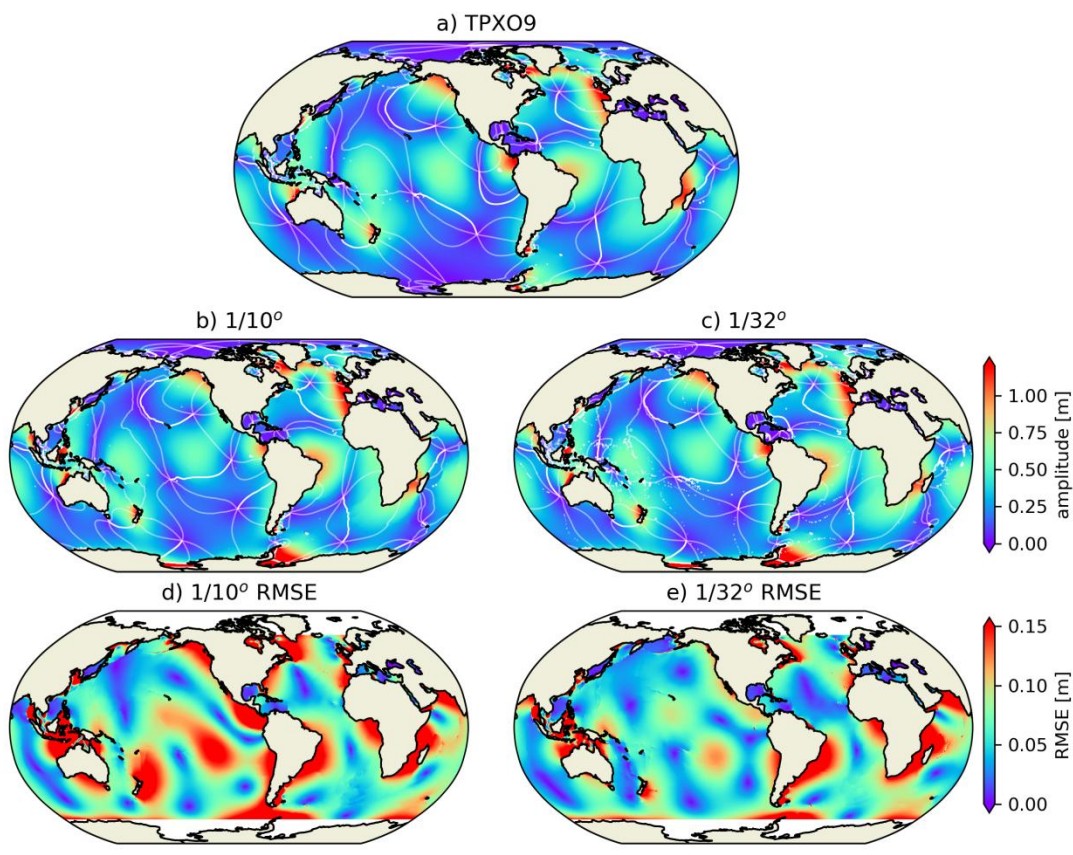

**Figure 5 M₂ co-tidal charts of TPXO9 (a), barotropic tide of FIO-COM 1/10°(b) and 1/32°(c) model with topographic drag parameterisation best tuned and their RMSE errors (d and e)**



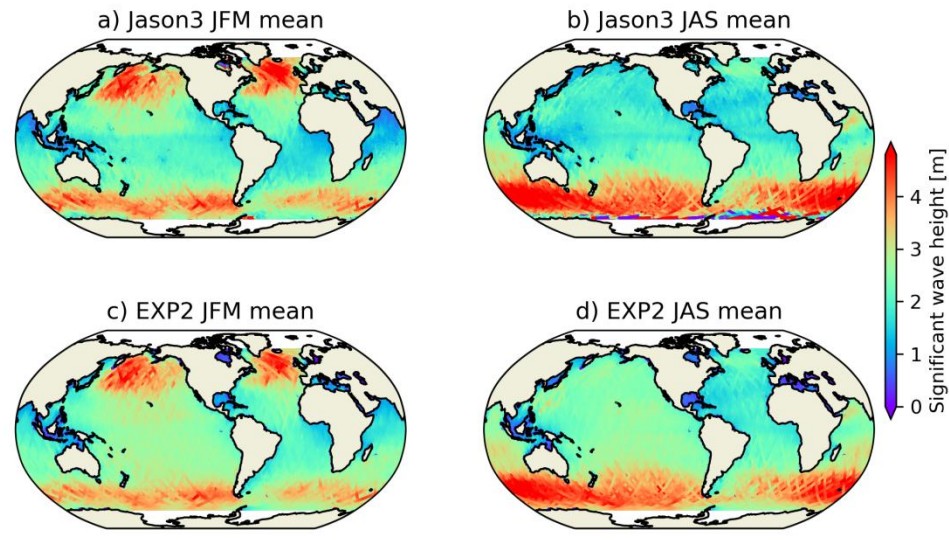

500           **Figure 6 Along track seasonal mean significant wave height of Jason3 (a, b) and EXP2 (c,d)**

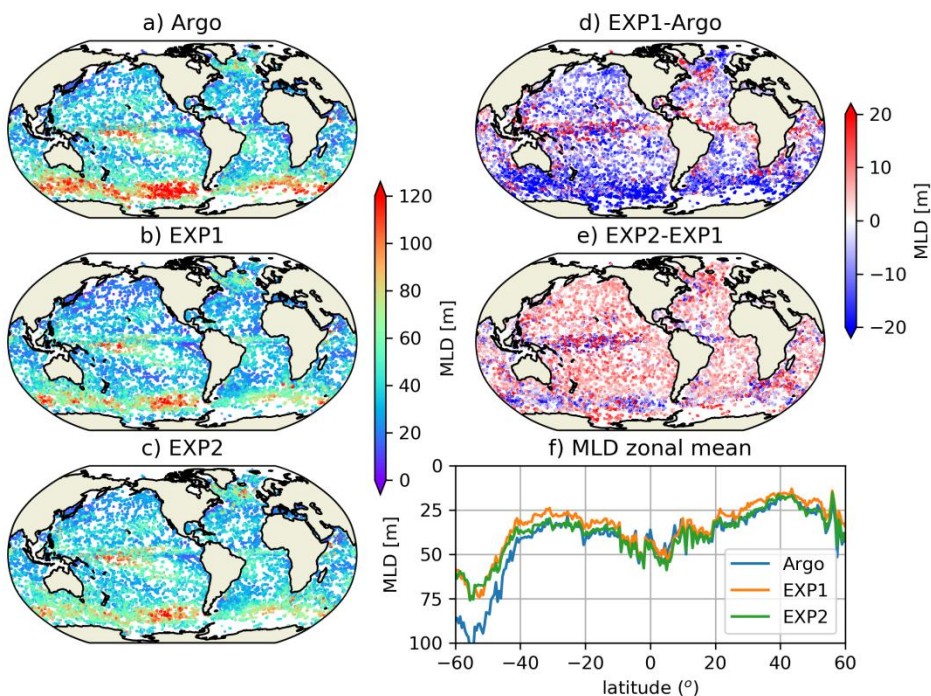

**Figure 7 Summer (JAS and JFM for northern and southern hemisphere, respectively) MLD based on Argo observations (a), EXP1 (b) and EXP2 (c) simulations. The differences of MLD between EXP1 and Argo, EXP2 and EXP1, and the zonal mean MLD are shown in (d), (e) and (f), respectively.**


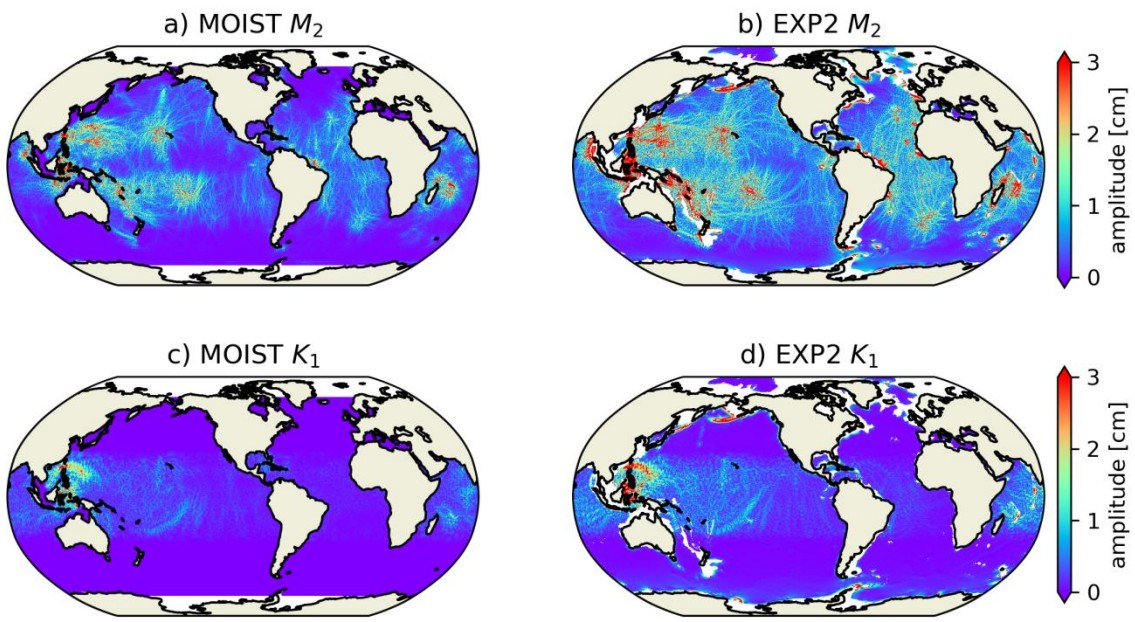

**Figure 8 M₂ internal tide amplitude of MOIST (a), EXP2 (b); (c), (d) same as (a), (b) but for K₁.**



**Figure 9 SSH wavenumber spectrum slope of 70-250km of Jason 3 along track data (a), EXP1 (b) and EXP2 (c). SSH wavenumber spectrum of three sites in regions of Kuroshio extension (d), tropical Pacific Ocean (e) and Southern Ocean (f). Snapshots of SSH of EXP1 (g) and EXP2 (h) on 1 December 2018 in the rectangle region shown in (a).**