# Peer review of "The development and validation of a global 1/32° surface wave-tidecirculation coupled ocean model: FIO-COM32"

_Geoscientific Model Development, 2022_

## Author Comment (AC1)

First of all, we all authors wish to thank Baylor Fox-Kemper and two anonymous referees for their insightful and constructive comments. We have carefully considered all comments and revised the manuscript accordingly. As response to the comments in GMDD, both the modifications those have already been done or are in progress are summarized as below, further response updates including lines of the paper where changes take place will be provided once all of the modifications are finished.

For the sake of clarity, referee comments are reproduced in blue colored fonts, and then we present our replies in black.

**Reviewer comments and our replies**

**RC1: 'Comment on gmd-2022-52', Baylor Fox-Kemper, 07 Apr 2022**

The paper "The development and validation of a global 1/32° surface wave-tide-circulation coupled ocean model: FIO-COM32" by Bin Xiao et al. describes the initial stages of developing a 1/32 degree version of the FIO model. Based on past successes of the FIO models, the Bv scheme is prominently discussed, as is the incorporation of tides. The paper focusses on the last year of two 2-3-year simulations EXP1, not including tides or Bv waves, and EXP2 including both. Two other important simulations are noted for future work.

In general, the paper describes a milestone of expensive work in progress, and for this reason many aspects of incomplete experimental design may be overlooked. However, some important theoretical aspects of the work are not mentioned (although they are relevant) and some additional analysis would be informative. Here is my short list of these issues:

1) In the mesoscale 1/10 degree model, wave effects on currents (WEC) and current effects on waves (CEW) are not expected to be very strong. However, as shown in McWilliams & Fox-Kemper (2013: http://dx.doi.org/10.1017/jfm.2013.348) and Suzuki et al. (2016: http://dx.doi.org/10.1002/2015JC011566) the expected magnitude of the WEC effects can be estimated using the epsilon parameter. Given the interest of FIO modeling to include

wave impacts in their modeling family, it would be very interesting to see the epsilon parameter estimated in the MASNUM-1/10 and MASNUM-1/32 models.

**Author response:** We have made estimation of the parameter $\epsilon$ to evaluate the Wave Effects on Currents (WEC), and added the related contents to the revised manuscript. The estimation is made for a typical section in Kuroshio extension region. We noticed that the major difference of the estimations between the 1/10° and 1/32° models is the frontal aspect ratio in $\epsilon$, which is $\frac{H}{L}$. During summer, the frontal aspect ratio is estimated to be ~1/1000 and ~3/1000 in 1/10° and 1/32° models respectively, which yield the estimated $\epsilon$ to be ~1/10 and ~3/10. While during winter, the frontal aspect ratio is estimated to be ~1/200 and ~3/200 in 1/10° and 1/32° models respectively, yielding the estimated $\epsilon$ to be ~1/2 and ~3/2. Hence, it can be concluded that as the horizontal resolution increased from 1/10° to 1/32°, the critical resolution may be reached at least during winter that the explicit stokes shear force becomes an important term. In this paper, the non-breaking wave induced mixing Bv is included and could be treated as a "bulk" mixing term accounting for these wave-turbulence interactions. It remain to be explored in the future the effect of explicit implementation of surface wave induced forces in the 1/32° model, and how they are compared with present numerical experiments.

2) It is not mentioned whether the EXP1 or EXP2 currents refract/diffract/affect the waves in the 1/32 models. It is well known from operational wave modeling that these effects become important roughly in the 1/10 resolution range. They are very important at 1/32 degree resolution. Some estimate of these effects would strengthen this work and provide impetus for a coupled wave-ocean simulation at this resolution to come.

**Author response:** We have added several new numerical experiments to test the Currents Effects on Waves (CEW). In the new CEW experiments, the surface currents of both the 1/10° and 1/32° of FIO-COM model are fed into the MASNUM wave model. We have got some preliminary results, further analysis and validations are in progress. We hope some robust estimations on the CEW can be made in time.

3) Given the offline MASNUM calculation, rather than the directly coupled MASNUM-1/32, it is probably impossible to include both the WEC and CEW effects in the

model. However, points 1&2 would show the need for such improvements. This is more interesting than the Bv parameterization result, which shows that small-scale turbulence parameterizations still affect simulations at this resolution. That is not surprising, given that those small-scale turbulence remain far below the resolution at 1/10, 1/32, and even 1/300 degree resolutions. What is more interesting as wave effects over the range of scales from 1/10 to 1/32 is the wave-current coupling.

**Author response:** We will address all of the suggested comments in the revised manuscript.

A) Aside from waves, the new information here primarily results from inclusion of tides. It is an interesting result that tides are significantly improved in the 1/32 degree over 1/10 degree model. However, most of the key metrics discuss only the coherent tides (e.g., Fig 5). As 1/32 degree current calculations could interact much more strongly with tides than the 1/10 degree model, some mention of enhanced incoherent tides would be interesting (and found from a straightforward comparison between EXP1 and EXP2).

**Author response:** The analysis of incoherent tides has been conducted, the corresponding modifications have been incorporated into the revised manuscript. A new figure is added showing the incoherent tide amplitude of semi-diurnal tidal band of 1/10° and 1/32° models. The incoherent tides is calculated as follows. Firstly, the coherent tide is obtained by applying harmonic analysis to the hourly SSH model output. Secondly, the incoherent signals are calculated by extracting the predicted coherent signals. Finally, a Butterworth 10th order band pass filter with semi-diurnal (1.73–2.13 cpd) is adopted to calculate the incoherent semi-diurnal tide time series. The incoherent tide amplitude is defined as the standard deviation of the incoherent tide time series. The incoherent tide amplitude of the 1/32° model is obviously stronger than that of 1/10° model. The increased incoherent tide amplitude in the 1/32° model should be attribute to the increased eddy activities and enhanced tide-circulation interactions.

B) There is no discussion of the subgrid damping used and how it scales with resolution. Furthermore, a power spectrum showing the rotational and divergent power spectra contributions would be extremely valuable in understanding how the 1/10 and 1/32 models differ at small scales. This information together with more information about the damping would be valuable in understanding the choices made and their consquences, as well

as the effective resolution of the vortical and wave/tide modes. This could supplement Fig 9 in a meaningful way, revealing more of the dynamics underpinning the better match of EXP2 to Jason than EXP1.

**Author response:** A discussion of the sub-grid damping will be added. A power spectrum showing the rotational and divergent power spectra contributions is now in progress. Further analysis on this topic will be conducted and added in the new version.

i) For a submesoscale-permitting model, it would be nice to see what submesoscales are expected to be permitted at 1/32 resolution. The stronger submesoscales in wintertime are now customary, but the weaker submesoscales in summer may be illustrating the limits of 1/32 resolution. It would be nice to include a discussion of Dong et al. (2020: http://dx.doi.org/10.1175/JPO-D-20-0043.1), along with some estimation from the MLD analysis as to the scale of submesoscale baroclinic instabilities. It would be particularly interesting to know if the Bv scheme deepens the MLD or mixes the stratification of the ML enough to have a detectable effect on MLI scale and whether it is more resolvable using Bv. Dong et al. has a similar analysis comparing MLI scales under different boundary layer schemes.

**Author response:** We are conducting further analysis on the mixed layer instability (MLI) following Dong et al. (2020), along with the estimation from the MLD analysis to investigate the scale of sub-mesoscale baroclinic instabilities. The effect of Bv scheme on summer MLI will also be investigated in the new version.

**RC2: 'Comment on gmd-2022-52', Anonymous Referee #2, 19 Apr 2022**

This paper summarizes two high-resolution experiments at 1/32 degree with and without tides. The simulations are spun up for 3 years and diagnostics are performed over the last two years.

This is a major computational exercise and the results are worth documenting. However, as presented, most of the results are expected and in agreement with previous studies. Perhaps the most novel result is the outcome of adding tides which leads to a better agreement between satellite-measured and modeled power spectra. It is indeed nice to see that an

increase in resolution leads to a better distribution and a higher magnitude of EKE, but for this paper to be publishable, it needs to go beyond a simple show and tell and provide new insights by performing more in depth analysis of the results and differences.

Specifically,

- What does 1/32 gives you in terms of which physical processes are better resolved? It is probably marginal in terms of the submesoscale, so do you see a difference in mixed layer instabilities between 1/10 and 1/32? Is the increase of EKE because of a stronger mesoscale field (lower viscosity) or the addition of submesoscale features?

**Author response:** As the horizontal resolution increases, small scale eddies and fronts are better resolved, these phenomenons are shown in Figure 3 in the old version. Besides, the representation of the large scale circulations, such as the path of the Kuroshio and Gulf Stream is also significantly improved (This will be explained latter). We have conducted new analysis to separate sub-mesoscale features from the larger scale motions to illustrate that both the stronger mesoscale field and addition of sub-mesoscale features are responsible for the increase of EKE. These analysis have been incorporated into the new version, and a new figure is added (Figure 2 in the new version). The analysis on mixed layer instabilities is in progress, and will be added into the revised manuscript.

- You allow for only one year spin up from the data assimilative 1/10 run. Is the KE in steady state? Is it sufficient for a mechanical adjustment?

**Author response:** We have added a new figure (Figure A - 1) in the appendix showing the time series of the total kinetic energy since the beginning of the numerical experiment. The time series shows that the 1/32° model restarts from a 1/10° model and enters a new steady state after an adjusting period of about 1 year. After the short spin-up period, the total kinetic energy manifest a quite steady evolution. This explanation has already been added to the revised manuscript.

- What is the T and S bias after 3 years? You do not use any relaxation to surface salinity which is known to lead to a significant drift in salinity. Can this be quantified? This

may not be the main focus of the paper, but it is of importance as it impacts the 3D T and S distribution and strength of the western boundary currents.

**Author response:** We have added a new figure (Figure A - 2) in the appendix showing model drift of temperature and salinity. Since the time span of the numerical experiments is not long (3.5 years), the model drift is generally small, even though we do not use any relaxation. The model drift of temperature shows a seasonal cycle in the upper ocean and a warming trend in the sub-surface layer, the maximum drift of temperature at the end is about 0.2℃. The upper 150 m becomes slightly salty and the sub-surface layer shows freshening drift, the salinity drift value is generally small, less than 0.02 PSU. This explanation has already been added to the revised manuscript. The analysis of strength of the western boundary currents is in progress, and will be added to the revised manuscript.

- This is more of a comment. You use relative wind which is known to have an eddy killing effect (Renault et al., 2019). This is reflected in a modeled EKE is lower than the smooth satellite observed geostrophic EKE.

**Author response:** Yes, we notice that the relative wind scheme may be responsible for the low EKE of the 1/10° model. In the future, we will test different wind stress schemes and their impacts in models with different resolutions.

- Line 62: What do you mean by "it is necessary to validate the effect of Bv"? What is the exact question being answered here? Is Bv still effective at 1/32? Do you have any reason what it should not? Is the impact of adding of Bv at 1/10 similar to that of 1/32? Is the mixed layer physic responding differently at 1/10 versus 1/32 (in other word, what is the impact on the MLD of resolving smaller oceanic features?)?

**Author response:** The exact questions what we want to propose are: Whether the model's MLD biases still exist in the high resolution 1/32° model, and what is the effect of Bv in the high resolution model. Previous works evaluating Bv effects were based on climatological runs and also validated models against climatological observations (e.g. Wang et al., 2019, doi: 10.1029/2018MS001494), how Bv affects the real-time atmosphere data forced numerical

experiments and validates against real-time observations? We have added these explanations to the revised manuscript.

Additional two numerical experiments with horizontal resolution of 1/10° is also conducted to investigate the effects of model resolution on simulated MLD. The settings of the two 1/10° model are kept identical with that EXP1 and EXP2 of 1/32° model respectively, hence they are named as EXP1Low and EXP2Low. We noticed that the impact of Bv in 1/10° model is quite similar to that of 1/32° model, the following figure shows the Bv effect in 1/10° experiments (EXP1Low and EXP2Low), which is quite similar as that of 1/32 (EXP1 and EXP2). The above experiments suggest that increasing horizontal resolution sorely does not solve the MLD biases. This explanation has been added to the revised manuscript.

[Figure]

Figure R-1 Summer of 2019 (JAS and JFM for northern and southern hemispheres, respectively) MLD based on Argo observations (a), EXP1Low (b) and EXP2Low (c) simulations. The differences of MLD between EXP1Low and Argo, EXP2Low and EXP1Low, and the zonal mean MLD are shown in (d), (e) and (f), respectively.

Reference:
Wang, S., Wang, Q., Shu, Q., Scholz, P., Lohmann, G., & Qiao, F. (2019). Improving the upper-ocean temperature in an ocean climate model (FESOM 1.4): Shortwave penetration versus mixing induced by nonbreaking surface waves. Journal of Advances in Modeling Earth Systems, 11, 545– 557. https://doi.org/10.1029/2018MS001494

- Snapshots are not representative of a solution. Improvements in western boundary current separation, extent and EKE need to be quantified by comparison to observations, not just stating that they qualitatively look better. Figure 1 and 2 are very small and it is really hard to see how the solutions differ, except for gross patterns.

**Author response:** We have added quantitative analysis and new figures (Figures 4 and 5) to the revised manuscript: In order to better understand the effects of model resolution on the large scale circulations, we focus on the Kuroshio and Gulf Stream regions. We propose the Integrated Circulation Route Errors (ICRE) as a quantitative criteria to assess the simulated paths of Kuroshio and Gulf Stream. The ICRE is calculated as follows. Firstly, the path of the western boundary current is defined as the contour edge of SSH at level of 0.16m. Secondly, the ICRE is calculated as the total misfit area between the contour edge of the model and CMEMS.

As the resolution increases from 1/10° to 1/32°, the simulated Kuroshio path is significantly improved (Figures 4), the ICRE is decreased from $3.01 \times 10^{11}$ m$^2$ to $1.73 \times 10^{11}$ m$^2$. The most notable improvement is that the 1/32° model is able to reproduce the Kuroshio large meanders, while the 1/10° model can not. The ICRE of Gulf Stream is decreased from $4.85 \times 10^{11}$ m$^2$ of 1/10° model to $3.27 \times 10^{11}$ m$^2$ of 1/32° model. We should note that for the comparisons of Gulf Stream region, the re-circulation part is masked out to focus on the simulation of main path of Gulf Stream. Compared to the 1/32° model, the 1/10° model fails to reproduce the deep penetration of the Gulf Stream into the Atlantic Ocean (Figure 5).

- What is the rationale for presenting barotropic tidal results? There is some improvements with the increase in resolution, but they are relatively small and not significantly better in the 1/32. Furthermore, since you are not using any drag in the 1/32, how is the RMSE of the barotropic tides when compared to TPXO? How does it compare to the barotropic simulation?

**Author response:** We have added discussions on this and modified the corresponding figure to include tidal results of EXP2 which does not use topography drag parameterization.

Reasonable global tide accuracy is an important prerequisite that tide and ocean general circulation could be coupled together, since tuning global barotropic tide can also yield better model topography settings, especially that some numerically unstable topography features can be fixed through this practice. Thus we think that presenting barotropic tidal results is relevant in this paper.

The simulated global tide in EXP2, which is a global baroclinic tide model, is shown in the modified figure. The overall pattern agree well with the TPXO9. As has been stated, the topography drag scheme is turned off in EXP2. As a result, the simulated tidal amplitude is significantly larger than the barotropic experiments (modified figures b, d) with topography drag optimally tuned, especially in Atlantic Ocean and the eastern Pacific. The global averaged tidal elevation RMSE is 16.1 cm almost double of the topography drag optimally tuned experiments. On the other hand, the tidal amplitude of EXP2 in the western Pacific region agree well with that of TPXO9, the amplitude bias and RMSE is much smaller than that of Atlantic Ocean. This phenomenon may indicate that a considerable portion of tidal energy conversion in the western Pacific region can be explicitly resolved in the EXP2. Since the goals of this study is focused on the investigation of tide-circulation coupled processes, we believe that the global tide accuracy is sufficient to support the that.

- Can you provide a quantitative measure of "your belief that the global tide accuracy is reasonable" (line 230)?

**Author response:** As has been stated in previous response, we have added discussion on this in the revised manuscript.

- Line 268 – The MOIST data are significantly weaker than the model. How does it compare to other published tidal global models or in-situ observations? This needs to be better quantified, even if the difference cannot be fully explained with the current set of experiments.

**Author response:** We will compare with other published models in the revised manuscript.

- Line 296: Showing figures with more "textures" is not very informative. Can you quantify how the internal tides signature affect the specific locations and why? How were the three locations chosen? I presume it is because of different surface internal tide signature, but this would benefit from a thorough discussion of how the internal tides modify the spectra at each location. BTW, location of site C is not shown in Figure 9a.

**Author response:** A new figure with Hovmöller diagram of SSH at western Pacific section will be added to the revised manuscript to show how the internal tides signature affect the specific locations.

Three locations are chosen for representing typical conditions: 1, High EKE and highly active internal tides; 2, High EKE but in-active internal tides; and 3, typical equatorial regions We have adjusted the locations to be of better representative. The explanation is added to the revised manuscript.

Corresponding Figure is modified to show location of site C.

Minor comments:

- Why do you have a maximum depth of 7000 m in the tidal simulation and only 5500 m in the non-tidal simulation?

**Author response:** This is because the set up works of the 1/32° model is started without explicit tide at first, and for this type of ocean model, the maximum depth of 5500 m is quite popular for legacy reasons. While for tidal simulations the model topography needs additional considerations, and a maximum depth of 7000 m can resolve most of the deep topography.

**RC3: 'Comment on gmd-2022-52', Anonymous Referee #3, 03 May 2022**

This manuscript describes the implementation and initial results of simulations using a very high-resolution (1/32°) global ocean model, including waves and tides. This is an exceptional effort and adds to a small handful of similar very high-resolution simulations of the ocean which have been undertaken to date. The paper describes the results of including

"mixing from non-breaking waves", and from the (surface) tides, which in turn generate internal tides. There is clearly merit in publishing some of the results, particularly those describing the tides and internal tides and their implications for comparisons with satellite spectra (i.e. the internal tides induce significant surface variability, leading to better agreement with satellite observations in regions of otherwise low variability), but I think a major revision would be needed first. This would be to address concerns about the "mixing from non-breaking waves", and also to include some further analysis to look at the evolution of the deeper ocean.

The main difficulty with the paper is the inclusion of the Bv mixing term and referring to this as "mixing by non-breaking waves", as I do not think that Bv represents such mixing. While the theory is discussed in Qiao et al. (2004), it is simpler to refer to Qiao et al. (2010, Ocean Dynamics 60: 1339-1355), in which a single monochromatic wave is considered in section 2.5. Prandtl theory stipulates that a diffusivity or vertical mixing rate can be specified from the wave average of the vertical velocity perturbation (w') and some mixing length based on the vertical displacement (l') of a fluid particle via the formulation <w'l'>, which is defined to be Bv in equation 35 (using slightly different nomenclature). They then choose l' (below equation 35, their l3w)) as the orbital vertical excursion due to the wave. It is clear that in this case, for instance, the vertical velocity will be a maximum when the particle is at its mean position (l'=0), and w' will be zero when the particle is at its maximum vertical position, etc. That is, w' and l' are 90° out of phase, or in quadrature, so that <w'l'> = 0. This would actually result from taking the w' as the vertical component of the wave orbital velocity as specified in their equation 34 (their u3w), in which an "i" imposes a 90° shift as compared with l'. Instead, the choice (between equations 34 and 35) is made to take w' to be directly proportional to, and in phase with, l', so that the resulting average <w'l'> is NON-zero. This choice is difficult to understand and means that Bv will represent an arbitrary mixing term which adds a potentially significant amount of mixing to the ocean near-surface, but which does not represent mixing by non-breaking waves.

Therefore, the paper should remove all reference to Bv as a mixing term due to "non-breaking waves". Ideally EXP2 should be re-run without the inclusion of Bv. If this is not possible, EXP2 could be used by simply saying that this includes "Bv as a mixing term", but without referring to this as being "mixing by non-breaking waves". In particular, the phrase "mixing by non-breaking waves" should not be included anywhere in this paper. In this context, I suggest that Equation 1 should be removed, and also those parts of Figure 7 which show the differences due to Bv (panels(c), (e) and the green line in panel (f)), as this appears to be an unphysical mixing term. Continuing to refer to Bv as "mixing by non-breaking waves" in papers such as this one will only serve to increase the confusion and misunderstanding over this term in the ocean modelling community.

**Author response:** Thank you so much for your detailed comments. The key point is that: w' and l' are 90° out of phase, or in quadrature, so that $<w'l'> = 0$, and then non-breaking wave can not generate additional turbulence. However, this traditional and classical point of view may be problematic due to the following reasons.

Firstly, the conclusion that w' and l' are 90° out of phase is from traditional wave theory which is from mathematical simplification. The simplification is based on mathematics, not physics. As we all know, from this simplification, the surface wave is irrotational. O M Phillips, as the founder of classical wave theory, pointed as early as 1961 that "Although the use of potential theory has been very successful in describing certain aspects of the dynamics of gravity waves, it is known that in a real fluid the motion can not be truly irrotational". In fact, our research was triggered by this sentence.

Secondly, solid observations from our group and other research groups have supported Bv. In 2016, based on in-situ observation we revealed the mechanism how wave-turbulence interaction enhance the background turbulence, in other word, surface wave does generate turbulence. And we confirmed this conclusion with laboratory experiments in a wave tank (Ma et al, 2022). The observation from Sutherland et al (2013), Paskyabi and Fer (2014) confirmed the Bv.

Thirdly, Bv is model independent and can dramatically improve all ocean and climate models. As we know, lack of mixing in the upper ocean has been a bottleneck for ocean and climate model development for more than half century. As a result, the simulated mixed layer depth in the upper ocean is too shallow, sea surface temperature (SST) is overheat and subsurface sea temperature is too low in summer for all ocean models. Since upper ocean plays key role in climate models, so all climate models have huge SST bias. By applying Bv, the simulation error can be reduced by about 80-90%, and climate modeled SST bias can be reduced about half. As Bv can really improve ocean and climate models, it has been employed in NEMO model, GFDL climate model (Fan and Griffies, 2014), and AWI FESOM model (Wang et al, 2019). All above models are leading models in the world.

Reference:

O. M. Phillips, 1961, A note on the turbulence generated by gravity waves. Journal of Geophysical Research, 66(9): 2889-2893.

Qiao F, Yuan Y, Deng J, Dai D, Song Z. 2016, Wave–turbulence interaction-induced vertical mixing and its effects in ocean and climate models. Phil. Trans. R. Soc. A 374:20150201. http://dx.doi.org/10.1098/rsta.2015.0201.

Ma H, Dai D, Jiang S, Huang C, Deng J, Qian F. 2022, Quantitatively study on wave-turbulence interactions by laboratory experiments. Dynam. Atmos. Oceans (In press).

Sutherland G, Ward B, Christensen K H, 2013, Wave-turbulence scaling in the ocean mixed layer. Ocean Science, 9(4): 597-608    DOI: 10.5194/os-9-597-2013.

Paskyabi M B, Fer I, 2014, The influence of surface gravity waves on the injection of turbulence in the upper ocean. Nonlinear processes in Geophysics, 21(3): 713-733    DOI: 10.5194/npg-21-713-2014.

Fan Y L, Griffies S M, 2014, Impacts of Parameterized Langmuir Turbulence and Nonbreaking Wave Mixing in Global Climate Simulations. Journal of Climate, 27(12): 4752-4775, DOI: 10.1175/JCLI-D-13-00583.1.

Wang S, Wang Q, Shu Q, Scholz P, Lohmann G, & Qiao F, 2019, Improving the upper-ocean temperature in an ocean climate model (FESOM 1.4): Shortwave penetration versus mixing induced by nonbreaking surface waves. Journal of Advances in Modeling Earth Systems, 11,545–557. https://doi.org/10.1029/2018MS001494.

On the other hand, I would like to see a more complete analysis of the behaviour of the model. In particular, it would be important to see how good the model is for purposes of climate modelling, for which the maintenance of a reasonably stable inventory of water masses is needed, or that the model should not drift too quickly from the initial conditions. I suggest the paper therefore include plots to show how the deeper water masses are drifting, and include figures showing the globally-averaged Temperature and Salinity (T and S) anomalies (differences from initial conditions) versus depth and time, also zonally-averaged (or sections at say 40°W and in middle of Pacific) T and S versus depth and latitude at the end of the runs.

**Author response:** We have added a new figure (Figure A-2) in the appendix showing model drifts of temperature and salinity and zonal averaged temperature and salinity at the end of numerical experiment. Since the time span of the numerical experiments is not long (3.5 years), the model drift is generally not big, even though we do not use any relaxation. The model drift of temperature shows a seasonal cycle in the upper ocean and a warming weak trend in the sub-surface layer, the maximum drift of temperature at the end is about 0.2℃. The upper 150 m becomes slightly salty and the sub-surface layer shows freshening drift, the salinity drift value is generally small, less than 0.02 PSU. This explanation has already been added to the revised manuscript.

It would also be good to include a discussion about how the model scales on various numbers of processors e.g. a figure showing the run time for 1 year of simulation on various numbers of cores, if this is possible.

**Author response:** At present we have difficulty to show the run time of simulation on various numbers of cores, because the resources we use to do the numerical experiment in this paper is 150 nodes (4200 CPU cores), this is almost the minimum resources to run the 1/32° model. Since such kind of benchmark tests are necessary, they will be carried out when conditions permit in the future.

Further Comments

- The English is readable but would benefit from being checked by a native English speaker (e.g. in the Abstract alone in l.s 19, 23, etc).

**Author response:** The English in the revised manuscript will be checked by a native English speaker.

- l. 12 etc. FIO-COM is not a "fully coupled" or even a "coupled" surface wave-tide-circulation model (as claimed several times e.g. l. 12, l. 321) as the waves are being run offline and fed into the tide-circulation model. There is no coupling back from the ocean circulation (or tides) onto the wave field. These claims need to be moderated.

**Author response:** Thank you for make this point clear. Yes, the wave model is offline. In the wave model, we have the scheme for wave-current interaction, so it can be fully coupled easily. We will include this point in the revision.

- ls. 140-142: what are the barotropic and baroclinic experiments referred to here?

**Author response:** The barotropic experiments is the first step we take to set up a tide-circulation coupled model. In the barotropic experiments, the temperature and salinity is frozen and kept as constant to debug the simulated global barotropic tide. Reasonable global tide accuracy is an important prerequisite that tide and ocean general circulation could be coupled, and tuning the global barotropic tide can also yield better model topography settings, especially that some numerically unstable topography features can be fixed through this practice. The baroclinic experiments refer to the formal results of tide-circulation coupled ocean model (EXP1 and EXP2).

- ls. 162-164: was the viscosity higher in the 1/10° model than in the 1/32° model or the same (this is relevant for the EKE discussion, as it is usual to reduce the viscosity at higher resolution as more of the eddies are resolved).

**Author response:** The horizontal viscosity scheme is a bi-harmonic operator with diffusive velocity of 1.96 cm/s for momentum and 0.65 cm/s for tracers respectively. Both the 1/10° and 1/32° models use the same model settings. For the scheme we use, the viscosity is

calculated from diffusive velocity times the cube of the grid spacing, which means the viscosity is about 10 times smaller in the 1/32° model than that of 1/10° model. This explanation has been added to the revised manuscript.

- Fig. 1d. The caption should state that this is the globally-averaged EKE.

**Author response:** Thanks for the comments, this has been modified in the revised manuscript.

- ls. 188-190. Why is the 1/32° model EKE higher than in the satellite observations in Fig. 1d? Is this because the model can resolve the internal tides but the satellites do not have sufficient resolution to do so? What is the along-track resolution of the satellites, for instance?

**Author response:** We have conducted new analysis to separate sub-mesoscale motions from the larger scale motions (e.g. mesoscale) by applying a low-pass spatial filter with radius of ~50 km. Both the EKE of low-pass 1/32° model and CMEMS are on the same 1/4° grid which makes them more comparable. The time series of the EKE of the low-pass 1/32° model (the thick solid red line in added figure) is significant lower than that of original 1/32° model and also lower than CMEMS. The added analysis illustrates that addition of sub-mesoscale motions (such as small scale eddies and fronts) is responsible for that 1/32° model EKE is higher than in the satellite observations (Fig. 1d in initial manuscript). Since the 1/32° model used in this analysis and in the initial manuscript is based on the daily outputs of EXP1 which do not have explicit tide, it is not proved that internal tides have effects on the higher EKE. These explanation have been incorporated into the revised manuscript, and a new figure is added.

- Fig. 2. Titles on the subplots (a) and (b) says CMMES – this should be CMEMS.

**Author response:** This has been modified in the revised manuscript.

- ls. 213-222. This is mostly a description of the barotropic model set-up and it would be better to discuss this in section 2 (Model description) rather than here in the Results section.

**Author response:** This has been modified in the revised manuscript.

- ls. 233-240 and Fig. 6. It is not clear if the model results in panels (c) and (d) show the ¼° model as implied by the text, or the 1/32° model as implied by the figure caption.

**Author response:** Yes, ls. 233-240 and Fig. 6. panels (c) and (d) in the initial manuscript show results of a ¼° model as implied by the text. The figure caption has been modified in the revised manuscript.

- Fig. 8. Caption to say that these are internal tide amplitudes at the surface.

**Author response:** This has been modified in the revised manuscript.

- Fig. 8. The internal tides in EXP2 are more energetic than in the MOIST observations. This implies that the dissipation of the internal tides is not being properly handled. Please comment on this. Has any explicit dissipation been applied to the internal tides to reduce their propagation?

**Author response:** A proper dissipation scheme of the internal tide and its adaption with the traditional viscosity schemes that designed for circulation only models is daunting challenge itself. To our knowledge, there is no such kind of scheme that is ready to be implemented, this need to be explored in the future. While we do have conducted some preliminary explorations that is shown in the following figure. Two experiments based on the 1/10° model are conducted. The model settings are kept identical to the EXP2Low in the revision, except for their background vertical viscosity. Figure R-2b shows the $M_2$ internal tide amplitude of the experiment with a normal background vertical viscosity $1.0 \times 10^{-4}$ m$^2$/s, the simulated internal tides is similar to that shown in initial manuscripts, which is more energetic than the MOIST observations. While in the experiment with an inordinately large vertical viscosity of 1.0 m$^2$/s, the amplitude of the simulated internal tide is brought down to the comparable level of the MOIST observations. These preliminary explorations indicate that a proper scheme of vertical mixing considering the internal gravity wave dissipation in the tide-circulation coupled ocean model need to be developed. These explanations will be added to the revision.

[Figure]

Figure R-2 $M_2$ internal tide amplitude from steric SSH, MOIST (a), EXP2Low with background vertical viscosity 1.0 x 10$^{-4}$ m$^2$/s (b), EXP2Low with background vertical viscosity 1.0 x 10$^{-4}$ m$^2$/s (c) .

- Fig. 9. Add lines to panels (d) to (f) to show the 70-250 km wave number band referred to in the text (e.g. l. 279).

**Author response:** This has been modified in the revised manuscript.

- Fig. 9. Colour bar for panels (a) to (c) should show negative values i.e. from 0 to -5.4 (rather than from 0 to +5.4).

**Author response:** This has been modified in the revised manuscript.

- l.321 and elsewhere. The FIO-COM model is NOT fully coupled as the waves are run offline and fed into the tide-circulation model.

**Author response:** Thank you for make this point clear. Yes, the wave model is offline. In the wave model, we have the scheme for wave-current interaction, so it can be fully coupled easily. We will include this point in the revision.